# Synovial Fluid-Induced Aggregation Occurs across *Staphylococcus aureus* Clinical Isolates and is Mechanistically Independent of Attached Biofilm Formation

Amelia Staats,[a,b] Peter W. Burback,[b] Mostafa Eltobgy,[b] Dana M. Parker,[f] Amal O. Amer,[b] Daniel J. Wozniak,[a,b] Shu-Hua Wang,[e] Kurt B. Stevenson,[e] Kenneth L. Urish,[f] Paul Stoodley[b,c,d]

aDepartment of Microbiology, The Ohio State University, Columbus, Ohio, USA

bDepartment of Microbial Infection and Immunity, The Ohio State University, Columbus, Ohio, USA

cDepartment of Orthopaedics, The Ohio State University, Columbus, Ohio, USA

dNational Centre for Advanced Tribology at Southampton (nCATS), National Biofilm Innovation Centre (NBIC), Department of Mechanical Engineering, University of Southampton, United Kingdom

eDepartment of Internal Medicine, Division of Infectious Diseases, The Ohio State University Wexner Medical Center, Columbus, Ohio, USA

fArthritis and Arthroplasty Design Group, Department of Orthopaedic Surgery, University of Pittsburgh, Pittsburgh, Pennsylvania, USA

**ABSTRACT** Rapid synovial fluid-induced aggregation of *Staphylococcus aureus* is currently being investigated as an important factor in the establishment of periprosthetic joint infections (PJIs). Pathogenic advantages of aggregate formation have been well documented *in vitro*, including recalcitrance to antibiotics and protection from host immune defenses. The objective of the present work was to determine the strain dependency of synovial fluid-induced aggregation by measuring the degree of aggregation of 21 clinical *S. aureus* isolates cultured from either PJI or bloodstream infections using imaging and flow cytometry. Furthermore, by measuring attached bacterial biomass using a conventional crystal violet assay, we assessed whether there is a correlation between the aggregative phenotype and surface-associated biofilm formation. While all of the isolates were stimulated to aggregate upon exposure to bovine synovial fluid (BSF) and human serum (HS), the extent of aggregation was highly variable between individual strains. Interestingly, the PJI isolates aggregated significantly more upon BSF exposure than those isolated from bloodstream infections. While we were able to stimulate biofilm formation with all of the isolates in growth medium, supplementation with either synovial fluid or human serum inhibited bacterial surface attachment over a 24 h incubation. Surprisingly, there was no correlation between the degree of synovial fluid-induced aggregation and quantity of surface-associated biofilm as measured by a conventional biofilm assay without host fluid supplementation. Taken together, our findings suggest that synovial fluid-induced aggregation appears to be widespread among *S. aureus* strains and mechanistically independent of biofilm formation.

**IMPORTANCE** Bacterial infections of hip and knee implants are rare but devastating complications of orthopedic surgery. Despite a widespread appreciation of the considerable financial, physical, and emotional burden associated with the development of a prosthetic joint infection, the establishment of bacteria in the synovial joint remains poorly understood. It has been shown that immediately upon exposure to synovial fluid, the viscous fluid in the joint, *Staphylococcus aureus* rapidly forms aggregates which are resistant to antibiotics and host immune cell clearance. The bacterial virulence associated with aggregate formation is likely a step in the establishment of prosthetic joint infection, and as such, it has the potential to be a potent target of prevention. We hope that this work contributes to the future development

Address correspondence to Amelia Staats, staats.43@osu.edu.

of therapeutics targeting synovial fluid-induced aggregation to better prevent and treat these infections.

KEYWORDS biofilm, orthopedics, aggregation, joint, infection, *S. aureus*, *Staphylococcus aureus*

The incidence of periprosthetic joint infection (PJI) following orthopedic surgery has remained persistently at approximately 2% (1). While the rate of infection is relatively low, the drastic increase in the number of annual orthopedic procedures, and associated severity of chronic PJI, warrants an urgent need for new preventative measures and therapeutics (2, 3). In order to develop novel targeted and non-invasive treatment options, a comprehensive understanding of PJI manifestation is critical.

Chronic PJI is frequently associated with the establishment of a mature staphylococcal biofilm adhering to the prosthetic implant or adjacent tissue within the joint cavity (4, 5). Current treatment options for persistent PJI involving a surface-associated biofilm are arduous, often requiring surgical explantation, placing of antibiotic-impregnated cement joint spacers, and rigorous intravenous antibiotics (6). Furthermore, severe cases may require long-term suppressive antibiotic regimens (7). Despite the significant clinical burden associated with staphylococcal biofilm-implicated infections, the mechanism of bacterial establishment in the periprosthetic joint space remains elusive.

In order to form a mature biofilm, bacterial cells entering the surgical site must first survive numerous challenges posed in the harsh joint cavity (8). Upon initial entry, they are expected to activate multiple elements of the innate immune system, including the rapid recruitment of circulating neutrophils (9). Neutrophil-mediated killing, antibiotic administration and fluid shear stress within the periprosthetic joint all present an immediate threat to the bacterial intruders (2, 10, 11). While *Staphylococcus aureus* employs a plethora of surface-bound (microbial surface components recognizing adhesive matrix molecules [MSCRAMMs]; clumping factors, collagen adhesins, fibronectin and fibrinogen-binding proteins, elastin-binding, host cell adhesions) and secreted virulence factors (leukocidins, hyaluronidases, lipases, etc.) to combat such hazards, the role of rapid bacterial aggregation within the joint is under investigation as an early protective step in the mechanism of establishing chronic PJI (12–15).

Following prosthetic device implantation, synovial fluid, a viscous substance secreted by resident synoviocytes, reoccupies the joint cavity (16). Recent *in vitro* data show that immediately upon exposure to synovial fluid *in vitro*, *S. aureus* promptly aggregates into macroscopic clusters (17, 18). Within just 6 h of contact, aggregate formation confers bacterial protection from high concentrations of antibiotics (19). Furthermore, staphylococcal aggregates grown on a serum-coated surface for 3 h display protection from neutrophil phagocytosis, likely due to a physical size threshold (9, 20).

It has since been reported that staphylococcal aggregates are present in synovial fluid aspirates of patients with chronic joint infections (21). Moreover, fluorescence microscopy of the extracted aggregates revealed a composition of biofilm matrix components, including fibrin, polysaccharides, and extracellular DNA (21). It is evident that aggregate formation, which is at least partially facilitated by the binding of synovial fluid protein components, likely contributes to the pathogenesis of *S. aureus* in the context of chronic PJI (18). As such, it is important to elucidate the spectrum of strains and organisms which have the capacity to undergo this process as a mechanism of virulence.

Though significant progress has been made toward characterizing aggregate composition and describing the kinetics of their formation, the pathogen specificity and long-term implications of this phenomenon have yet to be explored. Using a flow cytometry-based technique, we sought to quantify aggregation across a collection of *S. aureus* clinical isolates following exposure to either bovine synovial fluid (BSF) or human serum, since blood also enters the surgical site during the procedure. BSF was chosen because it has been documented to stimulate comparable staphylococcal aggregates to human, equine, and porcine synovial fluid and we could obtain large

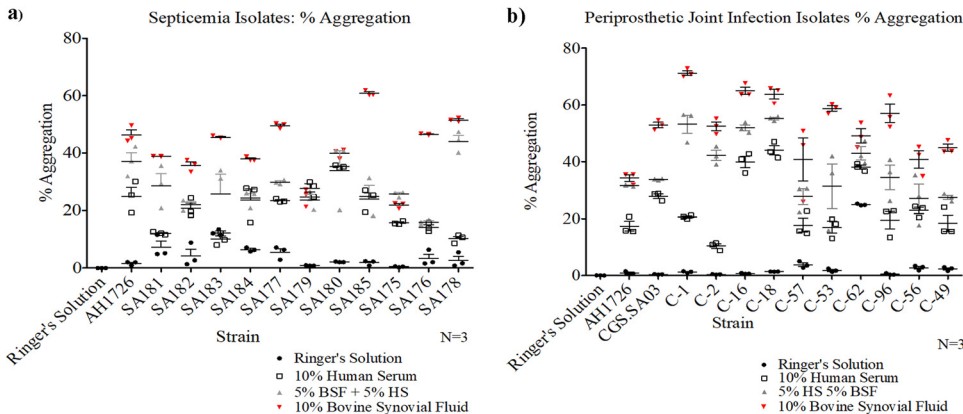

**FIG 1** Periprosthetic joint infection and septicemia *Staphylococcus aureus* clinical isolates aggregate in bovine synovial fluid (BSF) and/or human serum. (a and b) Septicemia (a) and periprosthetic joint infection (b) clinical isolates were exposed to 10% bovine synovial fluid supernatant (BSF) for 1 h prior to flow cytometry quantification. The percentage of the sample population within aggregates was calculated by subtracting the planktonic control gating from the total population. Three biological replicates were quantified for each strain. Error bars indicate mean ± SEM.

volumes with the same lot number with the intent of reducing variability (21, 22). First, the capability of 10 PJI isolates to aggregate in synovial fluid and human serum was confirmed. As a comparison, aggregation of 11 septicemia isolates was measured under the same conditions. Furthermore, with the use of a crystal violet assay, we assessed the influence of synovial fluid and serum on aggregation and surface-associated biofilm formation. Finally, we determined whether a correlation exists between host fluid aggregation and biofilm formation data.

## RESULTS

**Growth profiles of clinical isolates.** Prior to the aggregation experiments, 16 h growth curves were conducted for all of the PJI (see Fig. S1a in the supplemental material) and septicemia isolates (Fig. S1b). With the exception of PJI isolate C-62, which displayed a slower growth profile, all of the strains displayed similar growth curves over the course of a 16 h incubation in tryptic soy broth (TSB). Of note, C-62 also displayed a higher degree of auto-aggregation in Ringer's solution alone than the other strains (Fig. 1b), which may have influenced the optical density at 600 nm ($OD_{600}$) readings in the plate reader. This growth defect did not appear to influence the crystal violet biofilm assay of this strain.

To assess the influence of BSF and human serum on bacterial viability, we carried out CFU plating of our laboratory strain, AH1726, over the course of an 18 h exposure to either host fluid in TSB. After 1 h, there was no difference in bacterial viability between the untreated and BSF-supplemented cultures (Fig. S2a). While there was a temporary decline in CFU counts under the BSF-treated conditions at the second and third time points, by the final plating at 18 h, the culture had recovered to a similar concentration as the untreated bacteria. In contrast, when incubated with human serum in TSB, the treated and untreated cultures remained similar for the first 3 h, with a slight deficiency in the human serum-treated bacteria at the final 16 h time point (Fig. S2b).

In addition to our laboratory strain, the growth of a representative periprosthetic joint infection isolate and a septicemia isolate was assessed following incubation in both host fluids. Similar to AH1726, PJI isolate C-56 was inhibited by both the BSF (Fig. S2c) and the human serum (Fig. S2d) at the second and third time points but recovered by the final plating at hour 18. The same trends were observed in the septicemia isolate, OSUSA-175, when incubated in BSF (Fig. S2e) and human serum (Fig. S2f).

Finally, bacterial growth and viability were assessed in either human serum or BSF supplementation in Ringer's solution to evaluate the effect of the host fluids in a nutrient-free environment. For the first three time points, CFU counts of our laboratory strain

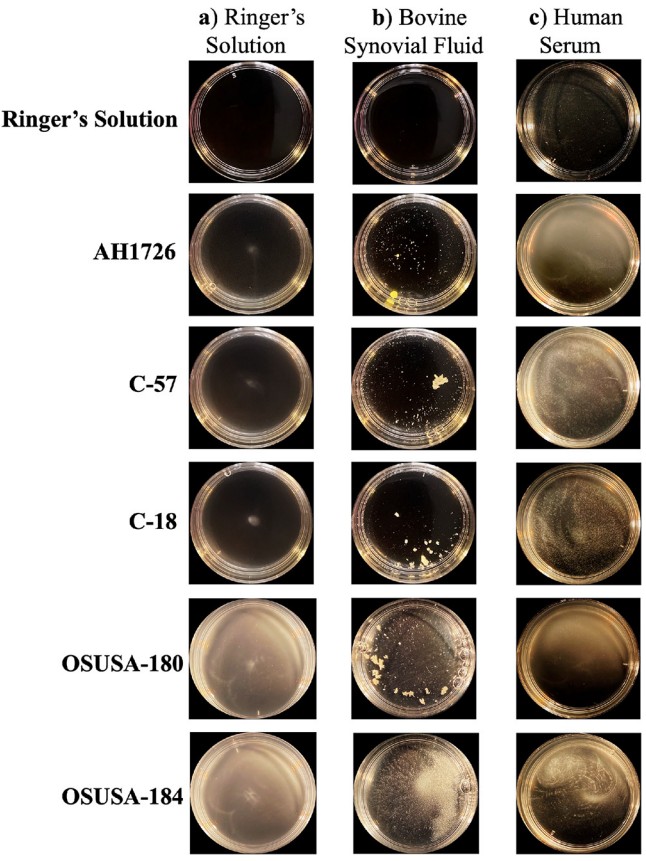

**FIG 2** Macroscopic images of synovial fluid-induced aggregation of clinical isolates. (a to c) Clinical isolates were incubated in Ringer's solution (R.S) (a), 10% bovine synovial fluid supernatant in R.S (b), or 10% human serum in R.S (c). Periprosthetic joint infection isolates are denoted by C-no. formatting, while septicemia isolates are denoted by OSUSA-no.

were comparable between the human serum-treated, BSF-treated, and untreated bacteria (Fig. S3a). Interestingly, by the final 18 h time point only the untreated bacteria had declined in bacterial viability, indicating that there may be some nutritional supplementation in the host fluids which can be utilized long term. However, based on these data, there was no growth enhancement during the 1 h aggregation assays resulting from host fluid inclusion. A 3 h viability plating was also conducted for the periprosthetic joint infection isolate, C-56 (Fig. S3b), and the septicemia isolate, OSUSA-175 (Fig. S3c), in Ringer's solution supplemented with host fluids. Both clinical isolates displayed counts comparable to those of the laboratory strain.

**Macroscopic imaging of isolates in synovial fluid and human serum.** The whole well of each clinical isolate was imaged following 1 h of incubation in Ringer's solution (Fig. 2a) or Ringer's solution supplemented with either BSF (Fig. 2b) or human serum (Fig. 2c). Macroscopically, we observed a diversity in both the degree of aggregation and external characteristics of the individual aggregates. While the BSF-stimulated aggregates were mostly large and globular, as represented by C-57, the human serum-stimulated aggregates were, in contrast, far smaller and scattered throughout the dish. This trend appeared to be conserved between the PJI and septicemia isolates, as well as laboratory strains. Two exceptions were septicemia isolates OSUSA-184 and OSUSA-175, which also displayed relatively low aggregation phenotypes as described by flow cytometry. Macroscopic imaging was also conducted for the remaining clinical isolates in human serum and bovine synovial fluid (Fig. S4).

**Synovial fluid-induced aggregation occurs across clinical isolates.** Using flow cytometry, the aggregation of 11 septicemia (Fig. 1a) and 10 PJI (Fig. 1b) *S. aureus*

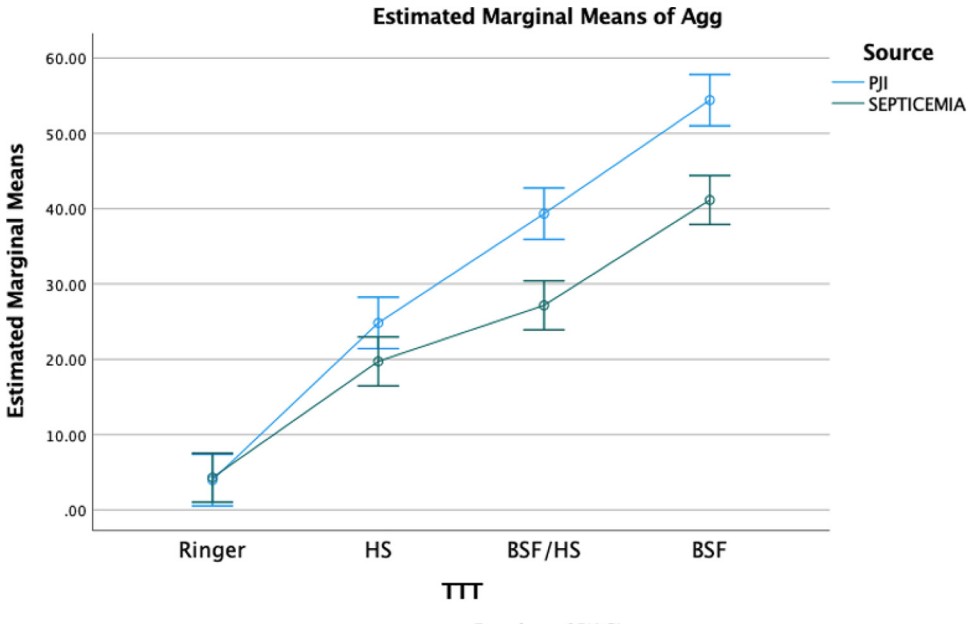

**FIG 3** Estimated marginal means of aggregation of PJI and septicemia isolates following a 1 h incubation in host fluids. Mean aggregation is plotted for each host fluid treatment (TTT). Error bars indicate 95% confidence interval.

clinical isolates was quantified. Following a 1 h incubation in Ringer's solution supplemented with BSF, the percentage of the sample population within aggregates was quantified. Bacteria were considered to be within an aggregate if the detected particle was larger than gated single cells and clusters within the planktonic control sample.

In Ringer's solution alone, only approximately 4% of the total PJI and septicemia isolates within a sample resided in an aggregate state. However, exposure to BSF increased aggregation to 54 and 41%, 13- and 10-fold increases, respectively ($P < 0.05$). Incubation in human serum resulted in the percentage of cells existing in aggregates to increase to 25 and 20% for the PJI and septicemia isolates, a significantly greater number than for Ringer's solution alone, but less than the BSF treatment. Furthermore, the mixture of BSF and human serum yielded a moderate degree of aggregation, with the percentage aggregation settling between the two mono-treatments. These data suggest that BSF has a greater aggregation potential than human serum.

A two-way analysis of variance (ANOVA) was conducted to examine both the source of infection (PJI versus septicemia) and host fluid treatment (BSF, human serum, or combination) as independent variables (Vs) and the percentage aggregation as the dependent variable (DV) (Fig. 3). We found that both the source of infection and the host fluid have a statistically significant effect on aggregation (Table 1). Our analysis revealed that only 14% of the variance in bacterial aggregation can be attributed to the source of infection (PJI versus septicemia) ($F$ [1,244] = 39.8; $P < 0.0001$; partial $\eta2 = 0.14$), while the host fluid treatment contributes 74% ($F$ [3,244] =236.5; $P < 0.0001$; partial $\eta2 = 0.74$). By Shapiro-wilk test, we determined that across most of the combinations of the independent variables, the aggregation values as a dependent variable were normally distributed. However, based on a series of Levene's $F$ tests, the homogeneity of variance assumption was not satisfied ($P < 0.05$). The violation of this assumption is not uncommon in two-way ANOVA; however, in order to minimize the risk of alpha error, a $P$ value of 0.0001 was adopted.

**PJI clinical isolates aggregate to a greater degree in synovial fluid than do septicemia isolates.** Although the extent of aggregation was similar between PJI and septicemia isolates on average, PJI isolates had 32% more cells in aggregates than the septicemia ($P < 0.05$) after incubation in BSF and 26% more in HS ($P < 0.05$) (Fig. 4a).

**TABLE 1** Two-way ANOVA pairwise comparisons in aggregation between host fluid treatments[a]

| Pairwise comparisons | | | | | | |
|---|---|---|---|---|---|---|
| **Dependent variable: aggregation** | | | | | **95% confidence interval for difference[c]** | |
| TTT | TTT | Mean difference[b] | SE | Sig.[b] | Lower bound | Upper bound |
| Ringer | HS | −18.145 | 1.693 | 0.000 | −22.648 | −13.643 |
| | BSF/HS | −29.120 | 1.693 | 0.000 | −33.622 | −24.618 |
| | BSF | −43.657 | 1.693 | 0.000 | −48.159 | −39.155 |
| HS | Ringer | 18.145 | 1.693 | 0.000 | 13.643 | 22.648 |
| | BSF/HS | −10.975 | 1.693 | 0.000 | −15.477 | −6.472 |
| | BSF | −25.512 | 1.693 | 0.000 | −30.014 | −21.009 |
| BSF/HS | Ringer | 29.120 | 1.693 | 0.000 | 24.618 | 33.622 |
| | HS | 10.975 | 1.693 | 0.000 | 6.472 | 15.477 |
| | BSF | −14.537 | 1.693 | 0.000 | −19.039 | −10.035 |
| BSF | Ringer | 43.657 | 1.693 | 0.000 | 39.155 | 48.159 |
| | HS | 25.512 | 1.693 | 0.000 | 21.009 | 30.014 |
| | BSF/HS | 14.537 | 1.693 | 0.000 | 10.035 | 19.039 |

[a]Based on estimated marginal means.
[b]The mean difference is significant at the 0.05 level.
[c]Adjustment for multiple comparisons, Bonferroni.

In contrast, there was no significant difference between the isolate groups when comparing surface-associated biofilm formation in polystyrene plates ($P < 0.05$) (Fig. 4b). A multivariate analysis of variance (MANOVA) was conducted to examine the source of infection (PJI versus septicemia) as an independent variable (IV) and both biofilm and synovial fluid-induced aggregation as dependent variables (DVs). Prior to conducting the MANOVA, normal distribution of the dependent variables was evaluated by Shapiro-Wilk test. Both variables' values were normally distributed ($P > 0.05$). Moreover, the homogeneity of variance assumption was tested. Based on a series of Levene's $F$ tests, the homogeneity of variance assumption for aggregation ($P = 0.8$) and biofilm ($P = 0.4$) was considered satisfied. Additionally, the equality of covariance matrices assumption was met ($P = 0.7$). MANOVA revealed nonsignificant associations between the biofilm formation and the source of infection (Table 1). However, the interaction between the aggregation and source of infection was significant ($F [1, 20] = 8.09$; $P < 0.01$; partial $\eta2 = 0.29$). Taken together, these data indicate that isolate

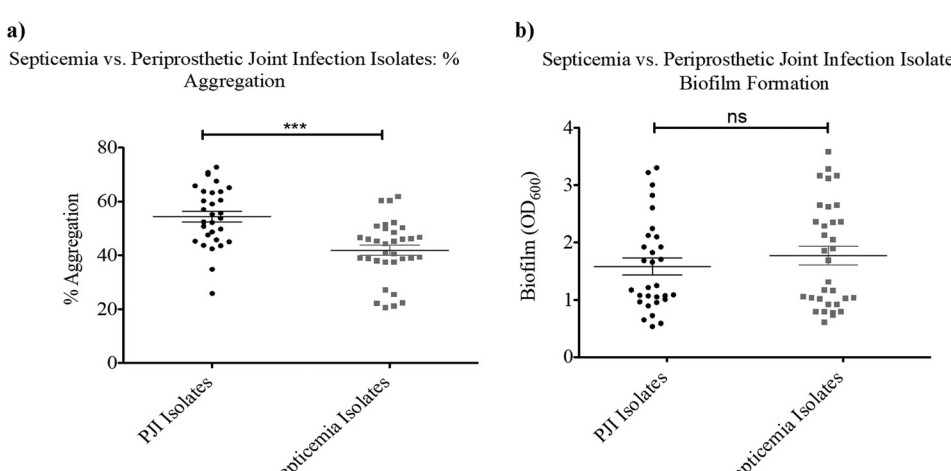

**FIG 4** Periprosthetic joint infection isolates aggregate significantly better in bovine synovial fluid than septicemia clinical isolates. There is no difference in attached biofilm formation between the two groups. (a and b) The average percentage aggregation (a) and attached biofilm formation (b) were calculated for 10 periprosthetic joint infection isolates and 11 septicemia isolates. Three biological replicates were included for each isolate. Error bars indicate the mean ± SEM. Statistical significance was determined by Student's $t$ test to compare means between treatments. ns (not significant), $P > 0.05$; ***, $P \leq 0.001$.

a) Septicemia Isolates Biofilm Formation

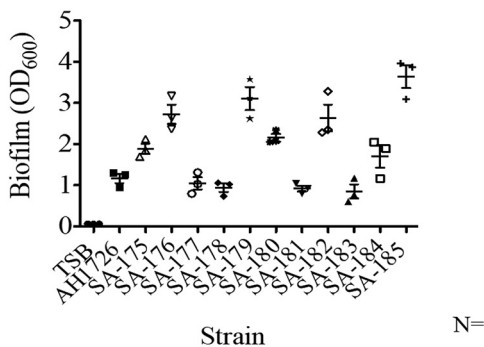

N=3

b)

Periprosthetic Joint Infection Isolates Biofilm Formation

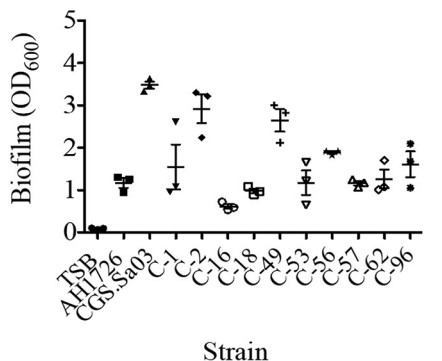

N=3

**FIG 5** Biofilm formation of septicemia and periprosthetic joint infection clinical isolates in tryptic soy broth (TSB). (a and b) Septicemia (a) and periprosthetic joint infection (b) clinical isolates were grown in TSB for 24 h prior to crystal violet staining and quantification of attached bacterial biofilm ($OD_{600}$). Three technical replicates were conducted for each strain. Error bars indicate the mean $\pm$ SEM.

source contributed to the variance in bacterial aggregation while having no effect on biofilm formation.

**Synovial fluid-induced aggregation does not correlate with attached biofilm formation in a conventional biofilm assay.** Following quantification of aggregation for each clinical isolate, crystal violet assays were conducted to quantify attached biofilm-forming capacity in the absence of BSF and human serum. All of the septicemia (Fig. 5a) and PJI isolates (Fig. 5b) formed attached biofilms within 24 h of incubation in a humidity chamber. While the extent of biofilm formation was highly variable between isolates, we found there to be no significant correlation between biofilm-forming capacity and synovial fluid-induced aggregation of the septicemia ($P = 0.8969$) (Fig. 6a) or PJI clinical isolates ($P = 0.5466$) (Fig. 6b, Table 2). In summary, the strains which produced greater surface-associated biofilm were not necessarily displaying high-aggregation phenotypes.

**Synovial fluid and human serum reduce bacterial surface attachment in polystyrene well plates.** To observe the impact of host fluids on surface-associated biofilm formation, we replicated the crystal violet assays with supplemented BSF, human serum, or a combination of both. Both our *S. aureus* laboratory strain, AH1726 (Fig. 7a), and clinical isolate, CGS.Sa03 (Fig. 7b), formed a biofilm in TSB alone. However, upon incubation with either host fluid, surface attachment and subsequent biofilm formation were significantly inhibited. While we clearly observed aggregate formation in the wells upon exposure to BSF and human serum, the aggregates were washed away with the rest of the planktonic bacteria during aspiration of the polystyrene well plates.

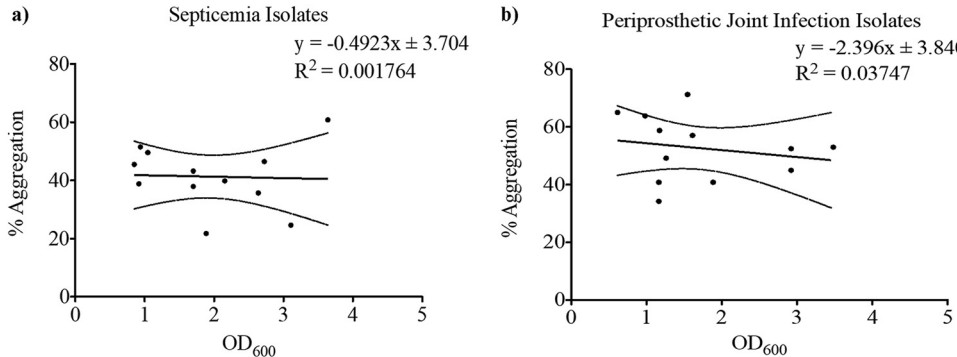

**FIG 6** There is no correlation between aggregation and surface-associated biofilm formation. For each septicemia isolate and PJI isolate, percentage aggregation and biofilm-forming capacity ($OD_{600}$) data were averaged, and a linear regression analysis was conducted. (a and b) No correlation was found between aggregation and biofilm formation for either septicemia isolates (a) or PJI Isolates (b) with $R^2$ values = 0.001764 and 0.03747, respectively.

## DISCUSSION

Synovial fluid-induced aggregation is under investigation as a powerful virulence factor aiding in early bacterial survival upon entry into the joint cavity. Previous reports demonstrate that the induction of *S. aureus* aggregates through synovial fluid exposure confers bacterial protection from high concentrations of antibiotics and immune cell clearance *in vitro* (9, 19, 20). Through this work, we have shown that both septicemia and PJI clinical isolates are stimulated to aggregate upon contact with BSF. These observations, along with previous studies demonstrating that synovial fluid-induced aggregation occurs with Gram-negative *Enterobacter* isolates, led us to conclude that a broad range of organisms display an aggregative phenotype (23).

In addition to synovial fluid, we also evaluated the influence of human serum on aggregate formation. While all of the isolates were stimulated to aggregate, it was to a lesser extent than in BSF. Moreover, when BSF was combined with human serum, the degree of aggregation fell between the two fluids alone. Our findings corroborate previously published work suggesting that rapid binding to fibrinogen and fibronectin are key elements in aggregate formation, both of which are abundant in synovial fluid but lacking in the serum component of blood (18). Of note, it has been previously documented that *S. aureus* expresses elevated levels of fibronectin-binding protein in synovial fluid compared with human serum (17). In contrast, expression of clumping factors was elevated in human serum compared to synovial fluid (17). These findings may explain why we observe different degrees of aggregation in BSF and human serum as well as distinct macroscopic aggregate morphologies.

Although all the isolates were stimulated to aggregate when exposed to BSF, strains isolated from PJIs aggregated significantly better than those isolated from bloodstream infection. While the dominating mechanisms of synovial fluid polymer binding are still under investigation, it is possible that there are genotypic differences between the strain groups which explain the enhanced aggregation observed in the PJI isolates. In recent work from Ma et al., 2020, it was observed that chronic PJI isolates yield a

**TABLE 2** Estimated marginal means in aggregation and biofilm formation between septicemia and PJI isolates

| Dependent variable | Source | Source | | 95% confidence interval | |
| | | Mean | SE | Lower bound | Upper bound |
|---|---|---|---|---|---|
| Biofilm | Septicemia | 1.942 | 0.250 | 1.421 | 2.463 |
| | PJI | 1.608 | 0.273 | 1.038 | 2.179 |
| Aggregation | Septicemia | 41.328 | 3.103 | 34.854 | 47.802 |
| | PJI | 54.420 | 3.400 | 47.328 | 61.512 |

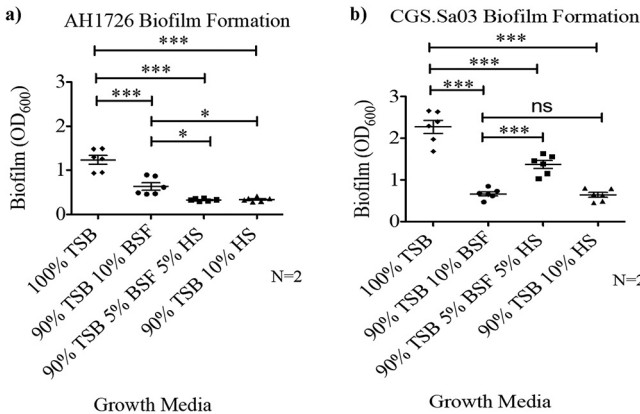

**FIG 7** Bovine synovial fluid (BSF) and human serum (HS) stimulate aggregation but inhibit the attachment step of surface-associated biofilm formation. Biofilm formation of *S. aureus* LAC strain AH1726 (Fig. 5a) and clinical isolate CGS.Sa03 (Fig. 5b) was quantified by a standard crystal violet assay. Biofilms were grown in tryptic soy broth (TSB), 10% BSF in TSB, 10% HS in TSB, or a combination treatment of 5% HS and 5% BSF in TSB. Two biological replicates consisting of three technical replicates were quantified for each strain. Error bars indicate the mean ± SEM. Statistical significance was determined by one-way ANOVA followed by Dunnett's multiple-comparison test to compare means between treatments. ns (not significant), $P > 0.05$; *, $P \leq 0.05$; **, $P \leq 0.01$; ***, $P \leq 0.001$.

significantly larger proportion of atypical strains than isolates cultured from the nares (24). Furthermore, over the course of a spreading infection, PJI isolates reportedly acquire mutations in genes associated with fibronectin-binding and -clumping factors (*clfA*, *clfB*, *fnbA*, and *ebh*). As the bacteria spread from the joint to adjacent host tissues, such adaptive changes likely influence progression of the infection (24). It is possible that these changes may contribute to both the aggregate and biofilm phenotypes of our isolates. Further characterization of the clinical strains will be assessed in future studies.

*S. aureus* expresses many surface adhesins and host component-binding factors which aid in attachment and subsequent biofilm proliferation (25). Expression of fibrinogen- and fibronectin-binding proteins allows for binding to synovial fluid protein components, thereby facilitating attachment to free-floating and/or surface-coating fibrinogen and fibronectin (26). We initially hypothesized that the ability of an isolate to bind synovial fluid components rapidly would lead to not only increased aggregation, but also surface attachment. We speculated that a higher aggregation phenotype would correlate with increased surface-associated biofilm formation with synovial fluid in the system. However, the addition of BSF or human serum to *S. aureus*, while stimulating the formation of aggregates, inhibited surface attachment and subsequent biofilm formation in a polystyrene plate. These results are consistent with published work reporting that the pretreatment of surfaces with synovial fluid inhibits *S. aureus* attachment (18). While the specific mechanics of attachment inhibition are unclear, there are a few possibilities which may explain our findings. It has been speculated that synovial fluid surface coating creates a protective barrier against bacterial attachment to implants through the formation of a conditioning film (18, 27). While *S. aureus* is capable of binding host proteins within the film such as fibrinogen and fibronectin, free-floating components may act as sequestering agents by way of blocking binding proteins and preventing additional bacterial interaction with the surface (18).

Using a conventional crystal violet assay in growth medium alone, we found no correlation between attached biofilm formation and synovial fluid-induced aggregation profile. These findings indicate that attached biofilm formation and synovial-fluid-induced aggregation are likely facilitated by independent mechanisms. While synovial fluid-induced aggregation is predominantly mediated by rapid host polymer binding,

surface attachment and biofilm formation, in contrast, are dominated by growth and exopolysaccharide (EPS) production during the developmental stages (10, 28).

To standardize our quantification of biofilm formation with previous studies, crystal violet assays were conducted in growth medium without supplementation of host fluids. However, our data suggest that the addition of relevant host fluids is important for assessing surface-associated biofilm formation of clinical isolates and extrapolating *in vitro* biofilm formation to the native environment (i.e., in our case, the joint space) and may explain inconsistencies with the use of a conventional biofilm assay (29).

The mechanism of biofilm formation in the joint cavity remains elusive. Whether synovial fluid-induced aggregation is a temporary state of protection upon initial entry or a preliminary step to aggregate-surface interaction is still unclear. As previously mentioned, recent studies report the presence of aggregates in the synovial fluid of patients with diagnosed joint infections (21). As such, it is probable that the long-term existence of synovial fluid-induced aggregates in the joint space may also be contributing to chronic infection independent of an attached biofilm. It is likely that aggregation is influenced by many other microbiological and host factors, including bacterial concentration, viscosity, and fluid dynamics. While understanding the relative contribution of each factor on synovial fluid-induced aggregation was beyond the scope of this study, we acknowledge the importance of their influence and are actively working to describe them in ongoing experiments.

Through the described work, we have shown that synovial fluid-induced aggregation is a widespread mechanism employed by distinct collections of staphylococcal strains. While all of the clinical isolates aggregated upon exposure to serum and synovial fluid, PJI isolates aggregated significantly more than septicemia isolates, suggesting genotypic differences between the groups. The supplementation of host fluids in a conventional crystal violet assay inhibited bacterial attachment in a well plate, suggesting that synovial fluid and/or serum coating may act as an inhibitory conditioning film, likely reflective of a native periprosthetic joint environment.

Finally, although all of the isolates formed biofilm to various degrees in growth medium alone, there was no correlation between biofilm-forming capacity and synovial fluid-induced aggregation. Of note, the use of polystyrene well plates for our biofilm assays may not represent bacterial attachment to a prosthetic implant or host tissue. We acknowledge this as a limitation of the present study and are working to assess synovial fluid-induced aggregate interaction with various orthopedic materials in ongoing experiments.

Taken together with our preliminary understanding of the molecular mechanism of aggregate formation in synovial fluid, we speculate that these two processes are independent of one another and likely cooccur in the context of biofilm-implicated PJI. While the role of host fluid-induced bacterial aggregation in the context of biofilm formation and chronic surgical site infection is still unclear, a wider screening for the aggregation phenotype in *S. aureus* isolates from various types of infections may reveal whether aggregation is an important, conserved virulence factor.

## MATERIALS AND METHODS

**Bacterial strains and growth conditions.** For all of the following experiments, *S. aureus* strains were streaked on tryptic soy agar (TSA) (BD Biosciences, Heidelberg, Germany) from frozen glycerol stocks and incubated at 37°C for 12 h to obtain isolated colonies. A total of 5 ml of tryptic soy broth (TSB) (BD Biosciences) was inoculated with a single colony and incubated for 12 h with shaking at 200 rpm at 37°C in an orbital shaker (Innova 44; New Brunswick Scientific). Green fluorescent protein (GFP)-expressing *S. aureus* LAC strain AH1726 was used as a laboratory strain in all assays along with the clinical isolates (18, 30). Additionally, *S. aureus* strain CGS.Sa03, isolated from an infected surgical mesh, was included in all assays, as it has been previously documented to both aggregate in synovial fluid and form biofilm in a standard crystal violet assay (18). PJI *S. aureus* strains were cultured from either synovial fluid or sonicated infected hardware at the University of Pittsburgh Department of Orthopaedic Surgery (Pittsburgh, PA). Ten of the PJI isolates (C-1, C-2, C-16, C-18, C-49, C-53, C-56, C-57, C-62, and C-96) were grown from frozen stocks on TSA plates for the following experiments (Table 3). Eleven *S. aureus* septicemia isolates (OSUSA-175, OSUSA-176, OSUSA-177, OSUSA-178, OSUSA-179, OSUSA-180, OSUSA-181, OSUSA-182, OSUSA-183, OSUSA-184, OSUSA-185) were cultured from bloodstream specimens at The

**TABLE 3** Origin of clinical isolates and details of infection

| Isolate[a] | Infection type | Isolate source | MRSA status[b] | Diagnosis (acute vs chronic) | Other |
|---|---|---|---|---|---|
| C-1 | Periprosthetic joint infection | Hip | + | Chronic | |
| C-2 | Periprosthetic joint infection | Hip | + | Chronic | |
| C-16 | Periprosthetic joint infection | Knee | + | Chronic | |
| C-18 | Periprosthetic joint infection | Knee | + | Chronic | |
| C-53 | Periprosthetic joint infection | Ankle | – (MSSA) | Chronic | Culture + for *Enterococcus faecalis* |
| C-62 | Periprosthetic joint infection | Knee | – (MSSA) | Chronic | |
| C-96 | Periprosthetic joint infection | Knee | + | Chronic | |
| C-56 | Periprosthetic joint infection | | – (MSSA) | | |
| C-49 | Periprosthetic joint infection | Knee | – (MSSA) | Acute | |
| C-57 | Periprosthetic joint infection | Knee | – (MSSA) | Acute | |
| SA175 | Bloodstream infection | Blood | + | Acute | Health care-associated |
| SA176 | Bloodstream infection | Blood | + | Acute | Health care-associated |
| SA177 | Bloodstream infection | Blood | + | Acute | Health care-associated community onset |
| SA178 | Bloodstream infection | Blood | + | Acute | Health care-associated community onset |
| SA179 | Bloodstream infection | Blood | + | Acute | Health care-associated community onset |
| SA180 | Bloodstream infection | Blood | + | Acute | Health care-associated community onset |
| SA181 | Bloodstream infection | Blood | + | Acute | Community-associated |
| SA182 | Bloodstream infection | Blood | + | Acute | Healthcare-associated |
| SA183 | Bloodstream infection | Blood | + | Acute | Health care-associated Community onset |
| SA184 | Bloodstream infection | Blood | + | Acute | Health care-associated community onset |
| SA185 | Bloodstream infection | Blood | + | Acute | Health care-associated community onset |

[a]C-1 and C-2 were isolated from same patient; C-16 and C-18 were isolated from same patient.
[b]MSSA, methicillin-sensitive *S. aureus*; MRSA, methicillin-resistant *S. aureus*. + indicates the strain tested positive for methicillin resistance; – indicates the strain tested negative for methicillin resistance but is methicillin sensitive.

Ohio State University Wexner Medical Center (Columbus, Ohio) (Table 3) (31). Clinical isolates were originally collected and provided to investigators for biofilm testing within a protocol approved by the institutions' institutional review boards (IRB).

**Growth curves.** To determine whether differences in aggregation or biofilm formation might be attributed to differences in growth, we measured planktonic growth rates. The 12 h overnight cultures were diluted 1:1,000 in TSB, and 200 $\mu$l of each dilution was subsequently transferred to 3 wells of a 96-well polystyrene plate for triplicate measurements (Thermo Fisher Scientific, Massachusetts, USA). The 16 h growth curves were conducted for each strain by measuring the optical density ($OD_{600}$) every 10 min with a plate reader (SpectraMax i3x; Molecular Devices) (18). Additionally, 18 h growth curves with bovine synovial fluid (BSF) (Lampire Biological Laboratories, Pipersville, PA, USA) or human serum (Sigma-Aldrich, St. Louis, MO, USA) in TSB were conducted to assess how much aggregation and biofilm formation may be attributed to physical interaction of the bacterial cells compared to growth. Finally, 3 h growth curves were conducted with either human serum or BSF in Ringer's solution (BR0052G; Fisher Scientific) to confirm that the addition of host fluids was not affecting bacterial viability during the 1 h aggregation assays. Growth curves with 10% BSF or 10% human serum diluted in either Ringer's solution or TSB were conducted by breaking up the aggregates with a syringe prior to sampling and plating on TSA for determining CFU counts. Plating was conducted every hour for 3 h, and then the cultures were allowed to incubate overnight for a final 18 h time point.

**Macroscopic imaging of bacterial aggregates.** Overnight cultures were diluted to an $OD_{600}$ of 0.5 and pelleted at 21,000 × *g* for 1 min prior to resuspension in 1 ml of Ringer's solution Each suspension was transferred to 35 by 10-mm petri dishes before adding an additional 1.7 ml of Ringer's solution. A subsequent aliquot of 300 $\mu$l of either BSF, PBS, or human serum was transferred to the dishes for a total volume of 3 ml. The cells were incubated for 1 h at 22°C with shaking at 60 rpm. Following incubation, a halogen light source illuminated the dishes for imaging against a black background. A dual 12-megapixel camera captured the images from 15 cm above the specimen.

**Flow cytometry-based quantification of bacterial aggregation.** All overnight cultures were normalized to an $OD_{600}$ of 0.75. Then, 250 $\mu$l of diluted cells were pelleted at 21,000 × *g* for 1 min and suspended in 250 $\mu$l of Ringer's solution as previously described (18). Cells were stained with SYTO-9 nucleic acid dye (Invitrogen, Thermo Fisher Scientific, Waltham MA, USA) for 10 min in order to differentiate bacteria from extraneous BSF components. Following staining, cells were washed twice with Ringer's solution to remove excess dye before resuspension in either Ringer's solution, 10% BSF in Ringer's solution, 10% human serum in Ringer's solution, or a combination of 5% BSF supernatant and 5% human serum in Ringer's solution. Prior to addition, BSF was centrifuged for 1 min at 21,000 × *g* to separate out tissue components as previously described (23). BSF supernatant was used for all assays (23). Cells were incubated in the respective host fluid treatments for 1 h, stagnantly at 22°C. Then, 100 $\mu$l of cells was gently transferred from the bottom of the tube into a 5-ml polystyrene tube for flow cytometry quantification using a BD FACsCanto flow cytometer (BD Biosciences, Franklin Lakes, NJ). A control sample of cells incubated in Ringer's solution only was gated to allow for a differentiation between planktonic bacteria (single cells and small clusters) and aggregates (events outside the

planktonic gate). Quantification of the forward and side scatter was conducted for the SYTO9+ population, ensuring that only bacteria was measured. The percentage of the sample population residing within aggregates was calculated by subtracting events in the planktonic gating from the total population using FlowJo 9.0 software. Statistical significance was determined by Student's $t$ test or two-way ANOVA.

**Crystal violet assay for biofilm quantification.** For quantification of attached biofilm, a crystal violet assay was adapted from Ma et al. (32). Overnight *S. aureus* cultures were grown as described above for 12 h with shaking at 200 rpm in a 37°C incubator. Then, 50 $\mu$l of overnight culture was used to inoculate 5 ml of TSB. The day culture was grown for 3 to 4 h with shaking at 200 rpm at 37°C to an $OD_{600}$ of 0.5. Next, 200 $\mu$l of day culture for each strain was added to 3 wells of a 96-well polystyrene plate. For experiments with BSF, human serum, or a mixture of both, 200 $\mu$l of culture was pelleted at 21,000 × $g$ for 1 min and suspended in 10% BSF supernatant, 10% human serum, or 5% of each in TSB. The resuspension was then added to the wells of the 96-well polystyrene plate. The plate was incubated in a humidity chamber (Styrofoam box coated with dampened towels) at 37°C for 24 h. Following incubation, wells were vacuum aspirated to remove unadhered cells and washed gently 3 times with phosphate-buffered saline (PBS). Then, 200 $\mu$l of 0.1% crystal violet stain (Sigma-Aldrich, St. Louis, MO, USA) was added to each well and allowed to incubate with the biofilm for 30 min. Following incubation, crystal violet was gently removed from wells, yielding stained bacterial biofilms. Wells were washed 3 times with PBS to remove excess dye prior to the addition of 200 $\mu$l of 33% glacial acetic acid (Fisher Scientific, Fisher Chemicals). Glacial acetic acid was incubated with the biofilm for 20 min for solubilization to occur. Finally, the $OD_{600}$ was measured using a plate reader (SpectraMax i3x; Molecular Devices, San Jose, CA, USA). Statistical significance was determined by unpaired Student's $t$ test using GraphPad Prism v5.0b software.

**Statistical analysis.** GraphPad Prism v5.0b software was used for statistical analysis of the following data. The threshold significance was set at a $P$ value of <0.05. All error bars indicate the standard error of the mean (SEM). SPSS Statistics was used for conducting univariate and multivariate analyses. A two-way ANOVA was conducted to determine the relative contribution of strain source (PJI versus septicemia) and fluid treatment (BSF and human serum) to the observed variation in aggregation. Furthermore, an unpaired, two-tailed Student's $t$ test was used to assess significant differences in biofilm formation within various host fluid treatments. A multivariate analysis of variance (MANOVA) was computed to determine if the source of infection explains a statistically significant amount of variance in synovial fluid-induced aggregation and attached biofilm formation.

## SUPPLEMENTAL MATERIAL

Supplemental material is available online only.

**SUPPLEMENTAL FILE 1**, PDF file, 0.8 MB.

## ACKNOWLEDGMENTS

We gratefully acknowledge NIH R01 GM124436 (P.S.), R01AI134895 and R01AI143916 (D.J.W.), A123121 and HL127651-01A1 (A.O.A.), and NIAMS R03AR077602 and K08 AR071494 (K.L.U.).

We declare no conflicts of interest.

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
