## [Reviewer comments · Microbiology Spectrum]

**Microbiology
Spectrum**

Synovial fluid-induced aggregation occurs across *Staphylococcus aureus* clinical isolates and is mechanistically independent of attached biofilm formation

Amelia Staats, Peter Burbach, Mostafa Eltobgy, Dana Parker, Amal Amer, Daniel Wozniak, Shu-Hua Wang, Kurt B. Stevenson, Kenneth Urish, and Paul Stoodley

Corresponding Author(s): Amelia Staats, The Ohio State University

Review Timeline:

Submission Date:	May 18, 2021
Editorial Decision:	July 12, 2021
Revision Received:	August 16, 2021
Accepted:	August 17, 2021

Editor: Kunyan Zhang

Reviewer(s): The reviewers have opted to remain anonymous.

Transaction Report:

DOI: <https://doi.org/10.1128/Spectrum.00267-21>

June 25, 2021

Amelia Staats
The Ohio State University
Department of Microbiology
460 W 12th Ave
Columbus, OH 43210

Re: Spectrum00267-21 (Synovial fluid-induced aggregation occurs across *Staphylococcus aureus* clinical isolates and is mechanistically independent of attached biofilm formation)

Dear Amelia Staats:

Thank you for submitting your manuscript to Microbiology Spectrum.

I seriously share the concern of Reviewer #2, regarding the similarity/no novelty of your results reported in current manuscript with your lab's previous publication [Pesttrak, M. J., et al. (2020), Investigation of synovial fluid induced *Staphylococcus aureus* aggregate development and its impact on surface attachment and biofilm formation. PLoS ONE, 15(4), e0231791]. My decision for publication of your manuscript will depend mainly on your clarification and justification of this concern.

When submitting the revised version of your paper, please provide (1) point-by-point responses to the issues raised by the reviewers as file type "Response to Reviewers," not in your cover letter, and (2) a PDF file that indicates the changes from the original submission (by highlighting or underlining the changes) as file type "Marked Up Manuscript - For Review Only". Please use this link to submit your revised manuscript - we strongly recommend that you submit your paper within the next 60 days or reach out to me. Detailed information on submitting your revised paper are below.

Link Not Available

Sincerely,

Kunyan Zhang

Journals Department
American Society for Microbiology
1752 N St., NW

Reviewer comments:

Reviewer #1 (Comments for the Author):

Recent studies have shown that *S. aureus* aggregates rapidly when exposed to the synovial fluid that is found in joints. However, whether aggregation of *S. aureus* is essential for establishing a prosthetic joint infection (PJI) and whether such behavior is qualitatively different from the ability of *S. aureus* to form a biofilm has not been rigorously tested. In this study, Staats et al. sought to understand these two different questions. The approach they utilized was to use a series of clinical *S. aureus* isolates of PJI and bloodstream origin and test whether the PJI isolates had adaptations that allowed them to aggregate better compared to the bloodstream isolates. After performing a series of controls and characterization of these clinical isolates for growth, colony counts, macroscopic aggregation, etc., the authors demonstrated that while all clinical isolates could aggregate, the PJI isolates had a significantly higher frequency of aggregates. This finding suggests that aggregation is likely to be a beneficial feature selected in *S. aureus* PJI isolates.

As for biofilm formation and how it may be related to aggregation, the authors found no correlation between these two phenotypes, which suggests that the mechanisms associated with adhering to plastic are different from intercellular adhesion seen during aggregation. This study is well-designed with appropriate controls and likely to be of some interest. The interpretation also seems robust; therefore, I do not have any substantial criticism of the data.

Some minor points that could be considered are as follows:

1. It is not clear to me what adaptations increase aggregation in PJI isolates. It would have been great to see if the expression of any known genes (eg. genes associated with fibrinogen and fibronectin binding proteins) involved in staphylococcal aggregation was up relative to internal control (sigB) in PJI isolates vs. septicemia isolates.
2. Although the authors have shown that biofilm formation on plastic does not correlate with aggregation, this is somewhat of a loose argument since biofilm formation depends on the available surface and coating on the surface. In this sense, the authors' results are somewhat obvious given their conditions to test biofilm formation.
3. The aggregation phenotypes in Fig 1 could easily be supplemental information given that it is quantified in Fig 2.
4. Axis titles need to be more prominent as it was difficult to read (esp., Fig 2.)

Reviewer #2 (Comments for the Author):

In the manuscript "Synovial fluid-induced aggregation occurs across *Staphylococcus aureus* clinical isolates and is mechanistically independent of attached biofilm formation", the authors determined the strain dependency of synovial fluid-induced aggregation. Furthermore, they assessed whether there is a correlation between the aggregative phenotype and surface-associated biofilm formation. For that, they have quantified the aggregation of *S. aureus* isolates from either PJI or bloodstream infections and determined the effect of synovial fluid (BSF) and human serum (HS) on *S. aureus* aggregation and biofilm formation. They found that: i) BSF and HS stimulated aggregation. ii) PJI isolates aggregated significantly more than *S. aureus* isolates from bloodstream infections. iii) BSF

and HS inhibited bacterial surface attachment. iv) There was no correlation between the degree of BSF aggregation and biofilm formation.

The text is clear and the experiments appear to be well performed. However, many of the results obtained are extremely descriptive and fully expected. Similar conclusions were obtained in a recent work from the same authors (Pesttrak, M. J., et al. (2020). Investigation of synovial fluid induced Staphylococcus aureus aggregate development and its impact on surface attachment and biofilm formation. PLoS ONE, 15(4), e0231791). Therefore, a more extensive study, including the molecular mechanisms that govern the observed phenotypes would strengthen the novelty and the conclusions of the work presented.

Specific comments

- Figure 1 is extremely large. It is recommended to reduce the size of the figure.
- The effect of BSF and HS on bacterial viability should be analyzed in a representative number of S. aureus isolates of each group and not only in the laboratory strain, AH1726.
- Since OSUSA-184 and OSUSA-175 isolates failed to aggregate it would be interesting to determine the molecular mechanisms by which these strains failed to aggregate. Did they express lower levels of fibronectin binding proteins?
- Since PJI isolates aggregate more efficiently than those isolated from bloodstream infections it would be interesting to determine if there are genotypic differences between the strains that would explain the differences in the aggregation phenotype.
- Different media are used for the experiments: macroscopic images of synovial fluid-induced aggregation of clinical isolates were performed in phosphate buffered saline (PBS); quantification of the aggregates by flow cytometry was performed using Ringer's solution; biofilm formation of septicemia and periprosthetic joint infection clinical isolates was performed using tryptic soy broth (TSB). I suggest the authors justify the selection of the different media for each analysis.

Reviewer #3 (Comments for the Author):

In "Synovial fluid-induced aggregation occurs across Staphylococcus aureus clinical isolates and is mechanistically independent of attached biofilm formation" the authors investigated synovial fluid-induced aggregation of clinical isolates of Staphylococcus aureus. The known synovial fluid-induced aggregation was confirmed with bovine synovial fluid. In addition, aggregation was also induced by human serum. Interestingly, bacterial surface attachment was inhibited by both bovine synovial fluid as well as by human serum. Taken together, the ability of bovine synovial fluid and human serum to induce aggregation while reducing surface attachment is highly relevant for improving the present concept of clinically relevant biofilm and aggregation. Even though the findings are interesting and important encouraging, the manuscript still raises concerns.

In general, the manuscript is too premature, and the structure needs to be improved. I have some suggestions.

In the "Importance":

Line 54-58 should be removed. They are too repetitive of the abstract. The remaining should be more focused on the impact of the findings.

In the "Materials and methods":

The information of the suppliers is often incomplete and inconsistent.

In the "Results":

The figures should be called for in the same sequence as they are numbered.

Don't present the supplementary figures in the beginning. The supplementary figures should follow the main figures as confirmative information.

In the "Figures":

The funds need to be increased

Figur 1 should go to the supplement and should be replaced by figure 2.

In the "Tables":

The funds need to be increased

The headings need to be improved

Staff Comments:

Preparing Revision Guidelines

For complete guidelines on revision requirements, please see the Instructions to Authors at [link to page]. **Submissions of a paper that does not conform to Microbiology Spectrum guidelines will delay acceptance of your manuscript.**

Please return the manuscript within 60 days; if you cannot complete the modification within this time period, please contact me. If you do not wish to modify the manuscript and prefer to submit it to another journal, please notify me of your decision immediately so that the manuscript may be formally withdrawn from consideration by Microbiology Spectrum.

If you would like to submit an image for consideration as the Featured Image for an issue, please contact Spectrum staff.

Response to Reviewers

Spectrum00267-21: Synovial fluid-induced aggregation occurs across *Staphylococcus aureus* clinical isolates and is mechanistically independent of attached biofilm formation

Editor Comments:

Comment 1: I seriously share the concern of Reviewer #2, regarding the similarity/no novelty of your results reported in current manuscript with your lab's previous publication [Pesttrak, M. J., et al. (2020), Investigation of synovial fluid induced *Staphylococcus aureus* aggregate development and its impact on surface attachment and biofilm formation. PLoS ONE, 15(4), e0231791].

Response 1:

We appreciate your time and comments regarding the review of our manuscript. I hope to provide some clarification of the concerns expressed. I've broken my response into two paragraphs, the first describing the purpose and findings of our previous work and the second focusing on how the present manuscript is distinct.

In the cited paper, [Pesttrak, M. J., et al. (2020), Investigation of synovial fluid induced *Staphylococcus aureus* aggregate development and its impact on surface attachment and biofilm formation. PLoS ONE, 15(4), e0231791], our group investigated the early kinetics of synovial fluid-induced aggregation using a single laboratory strain and a single clinical isolate. Through this study we showed a time-dependence on synovial fluid exposure, as well as reported the individual contributions of synovial fluid polymers to aggregation (fibrinogen, fibronectin, hyaluronic acid, extracellular DNA and BSA). Furthermore, in Pesttrak et al. a flow cell model was used in short time-lapse videos to assess flowing aggregate attachment to surfaces. Overall, the findings of this paper laid the groundwork for future mechanistic studies and established the flow cytometry-based quantification of aggregation as a quantitative methodology in our laboratory.

In contrast, the purpose of the present study was to determine how common synovial fluid-induced aggregation is amongst clinical strains which were isolated from prosthetic joint infection patients, using septicemia isolates as a comparator. To push our understanding of the clinical implications of this phenomenon, we conducted biofilm assays with host fluids in the system. In contrast with the work cited above, these experiments were carried out using conventional biofilm assays, conducted for 12 hours with growth media and host fluid present throughout the experiment. All previous work was confined to 5 minutes with synovial fluid diluted to negligible concentrations, primarily focusing on the early bacterial attachment step. Our current findings address the ability of the bacteria to aggregate, attach and form a biofilm as would be relevant in the context of chronic prosthetic joint infection—pushing our knowledge beyond 5 minutes of aggregate-surface interaction.

Respectfully, the purpose of this work was not to elucidate a molecular mechanism for synovial fluid-induced aggregation, but to further our knowledge of the clinical implications (biofilm

formation) and scope (strain-specificity) of staphylococcal synovial fluid-induced aggregation. Through these experiments we have uncovered meaningful trends, such as a high degree of variability in aggregation between strains and sources of isolates, which we feel will be beneficial in future studies aiming to elucidate this mechanism. However, these experiments will not be trivial due to the complexity of the system which consists of synovial fluid and the plethora of external factors which influence aggregation (fluid dynamics, viscosity, polymer concentrations). These experiments were beyond the scope of this work. We thank you very much for your consideration.

Reviewer 1 Comments

Comment 1. It is not clear to me what adaptations increase aggregation in PJI isolates. It would have been great to see if the expression of any known genes (eg. genes associated with fibrinogen and fibronectin binding proteins) involved in staphylococcal aggregation was up relative to internal control (sigB) in PJI isolates vs. septicemia isolates.

Response 1: We agree with this point and are currently working with the physicians from which we obtained the PJI and septicemia isolates to further this analysis. Genotyping of each isolate is currently being carried out to look at differences between the PJI and Septicemia groups. Furthermore, our lab is conducting RNA-sequencing experiments to discover gene expression which may be elevated upon contact with the synovial fluid. We hope these studies will contribute to a more mechanistic understanding this process, however, this was beyond the scope of the present work. We cannot definitively conclude that the adaptations discussed lead to increased aggregation in the PJI isolates. These adaptations were observed to occur as the infection spread from one point to another, indicating that they may play a role in the development of chronic infection. We have clarified this reference in the main text [**lines 363-369**].

Comment 2. Although the authors have shown that biofilm formation on plastic does not correlate with aggregation, this is somewhat of a loose argument since biofilm formation depends on the available surface and coating on the surface. In this sense, the authors' results are somewhat obvious given their conditions to test biofilm formation.

Response 2: We acknowledge this as a limitation of the study; however, crystal violet assays are a widely accepted, standardized method for quantifying biofilm formation of clinical isolates. Ongoing research in our lab is studying bacterial attachment to various orthopedic materials, as well as mapping biofilm on infected explants. While we agree it is entirely possible that the biofilm assay will be influenced by material type, for the purpose of this study and the inclusion of multiple clinical isolates in our screen, we opted for a well-documented and reproducible assay. Acknowledgment of this limitation has been included in the manuscript [**lines 422-425**]

Comment 3. The aggregation phenotypes in Fig 1 could easily be supplemental information given that it is quantified in Fig 2.

Response 3: Thank you for your recommendation. In addition to our laboratory strain, we have selected 2 representative strains from each isolate group to display the synovial fluid and human serum aggregation phenotype [line 635]. The remaining images will be moved supplemental [line 776].

Comment 4. Axis titles need to be more prominent as it was difficult to read (esp., Fig 2.)

Response 4: All figure texts have been increased.

Reviewer 2 Comments

Comment 1. However, many of the results obtained are extremely descriptive and fully expected. Similar conclusions were obtained in a recent work from the same authors (Pesttrak, M. J., et al. (2020). Investigation of synovial fluid induced *Staphylococcus aureus* aggregate development and its impact on surface attachment and biofilm formation. PLoS ONE, 15(4), e0231791). Therefore, a more extensive study, including the molecular mechanisms that govern the observed phenotypes would strengthen the novelty and the conclusions of the work presented.

Response 1: We agree that widespread synovial fluid-induced aggregation amongst the isolates was fully expected, however, we argue that our key findings, which were unanticipated, provide valuable, clinically relevant insight. For example, while all the isolates aggregated as expected upon synovial fluid exposure, there was a great deal of variability between them, suggesting there may be genotypic differences between strains contributing to this phenotype. Furthermore, PJI isolates aggregated significantly better than septicemia isolates. In future studies we plan to use these findings to delve into the molecular mechanisms which contribute to this variability. However, these experiments are not trivial due to the complexity of synovial fluid and the plethora of factors which influence aggregation (fluid dynamics, viscosity, polymer concentrations). Current work is being conducted, by our group and others, to better understand this process at the molecular level, but the purpose of the present study was to further our knowledge of the clinical implications and scope of staphylococcal synovial fluid-induced aggregation.

Comment 2. Figure 1 is extremely large. It is recommended to reduce the size of the figure.

Response 2: We appreciate the recommendation. We have two representative strains from each isolate group in addition to our laboratory strain (AH1726) to display the synovial fluid and human serum aggregation phenotype in the main text [line 635]. The rest of the images will move the rest to supplemental [line 776].

Comment 3. The effect of BSF and HS on bacterial viability should be analyzed in a representative number of *S. aureus* isolates of each group and not only in the laboratory strain, AH1726.

Response 3: We agree with this point and are now including bacterial viability tests for a representative PJI isolate as well as a septicemia isolate found in supplementary figure 2 [line 747] and supplementary figure 3 [line 757]. Similar host fluid effects were found for both clinical isolates compared to the laboratory strain.

Comment 4. Since OSUSA-184 and OSUSA-175 isolates failed to aggregate it would be interesting to determine the molecular mechanisms by which these strains failed to aggregate. Did they express lower levels of fibronectin binding proteins?

Response 4: Thank you for this suggestion. As mentioned previously this is ongoing work in the laboratory in collaboration with the surgical groups who provided the strains. Genotyping of all the isolates is being conducted to determine whether differences exist between the groups. As previously mentioned, our laboratory is also conducting RNA-seq experiments to quantify gene expression of these binding proteins following synovial fluid exposure. We hope to report these findings in the near future; however, they were beyond the purpose of the present work.

Comment 5. Since PJI isolates aggregate more efficiently than those isolated from bloodstream infections it would be interesting to determine if there were genotypic differences between the strains that would explain the differences in the aggregation phenotype.

Response 5: We agree with this point and are currently working with the physician groups from which we obtained the strains to conduct genotyping for future works. In addition to looking at genotypic differences between the PJI and septicemia isolates, we will also look at differences between high and low aggregators. Currently, we are working to identify specific bacterial (host-polymer binding proteins) which influence aggregation in synovial fluid and will guide our sequencing analysis. We are also actively collecting new PJI isolates which we will include in these future studies. x

Comment 6. Different media are used for the experiments: macroscopic images of synovial fluid-induced aggregation of clinical isolates were performed in phosphate buffered saline (PBS); quantification of the aggregates by flow cytometry was performed using Ringer's solution; biofilm formation of septicemia and periprosthetic joint infection clinical isolates was performed using tryptic soy broth (TSB). I suggest the authors justify the selection of the different media for each analysis.

Response 6: The use of two different saline solutions resulted from different people performing the experiments. As the macroscopic images were simply intended to be a descriptive display of the aggregate phenotype from our model, we did not repeat the imaging with Ringer's solution. Additionally, we have previously shown that PBS and Ringer's solution display the same aggregation profiles by flow cytometry and microscopy. However, we agree that consistency in our methodology is important, and imaging was repeated with Ringer's solution in the final manuscript [lines 158-159, 635, 776]. Furthermore, we repeated bacterial viability testing in

Ringer's solution which were previously conducted using PBS with host fluid supplementation [line 757].

For the biofilm assays, growth media is required to maintain bacteria viability overnight. These assays were intended to assess a distinct process from the short-term aggregate formation which requires a growth element.

Reviewer 3

Comment 1: In the "Importance": Line 54-58 should be removed. They are too repetitive of the abstract. The reaming should be more focused on the impact of the findings.

Response 1: Thank you for your recommendation, we have revised the importance section to focus on the broader impact of the findings [lines 50-56]. Furthermore, repetitive language was removed.

Comment 2: In the "Materials and methods": The information of the suppliers is often incomplete and inconsistent.

Response 2: We have thoroughly revised our materials and methods checked that all suppliers were reported [lines 121, 123, 124, 150, 177, 203].

Comment 3: In the "Figures": The fonts need to be increased

Response 3: Thank you for your comment, all fonts have been increased.

Comment 4: Figure 1 should go to the supplement and should be replaced by figure 2.

Response 4: We agree that figure one is too large for the main text, however, we believe it is useful to have the aggregate phenotype displayed. As mentioned previously, we have selected 5 representative strains to keep in the main text [line 635] with the rest moved to supplemental [line 776].

Comment 5: In the "Tables": The fonds need to be increased. The headings need to be improved.

Response 5: We have revised the headings to be more descriptive and increased the fonts.

August 17, 2021

Amelia Staats
The Ohio State University
Department of Microbiology
460 W 12th Ave
Columbus, OH 43210

Re: Spectrum00267-21R1 (Synovial fluid-induced aggregation occurs across *Staphylococcus aureus* clinical isolates and is mechanistically independent of attached biofilm formation)

Dear Amelia Staats:

Your manuscript has been accepted, and I am forwarding it to the ASM Journals Department for publication. You will be notified when your proofs are ready to be viewed.

Sincerely,

Kunyan Zhang
Editor, Microbiology Spectrum

Journals Department
Supplemental Material FOR publication: Accept